# Laminin Alpha 2 Enhances the Protective Effect of Exosomes on Human iPSC-Derived Cardiomyocytes in an In Vitro Ischemia-Reoxygenation Model

**DOI:** 10.3390/ijms25073773

**Published:** 2024-03-28

**Authors:** Fernanda C. P. Mesquita, Madelyn King, Patricia Luciana da Costa Lopez, Shiyanth Thevasagayampillai, Preethi H. Gunaratne, Camila Hochman-Mendez

**Affiliations:** 1Department of Regenerative Medicine Research, The Texas Heart Institute, Houston, TX 77030, USA; fmesquita@texasheart.org (F.C.P.M.); mking@texasheart.org (M.K.); plopez@texasheart.org (P.L.d.C.L.); 2Department of Biology and Biochemistry, University of Houston, Houston, TX 77204, USA

**Keywords:** ischemia-reperfusion injury, exosomes, laminin alpha 2, ischemia reoxygenation, cardiomyocytes

## Abstract

Ischemic heart disease, a leading cause of death worldwide, manifests clinically as myocardial infarction. Contemporary therapies using mesenchymal stromal cells (MSCs) and their derivative (exosomes, EXOs) were developed to decrease the progression of cell damage during ischemic injury. Laminin alpha 2 (LAMA2) is an important extracellular matrix protein of the heart. Here, we generated MSC-derived exosomes cultivated under LAMA2 coating to enhance human-induced pluripotent stem cell (hiPSC)-cardiomyocyte recognition of LAMA2-EXOs, thus, increasing cell protection during ischemia reoxygenation. We mapped the mRNA content of LAMA2 and gelatin-EXOs and identified 798 genes that were differentially expressed, including genes associated with cardiac muscle development and extracellular matrix organization. Cells were treated with LAMA2-EXOs 2 h before a 4 h ischemia period (1% O_2_, 5% CO_2_, glucose-free media). LAMA2-EXOs had a two-fold protective effect compared to non-treatment on plasma membrane integrity and the apoptosis activation pathway; after a 1.5 h recovery period (20% O_2_, 5% CO_2_, cardiomyocyte-enriched media), cardiomyocytes treated with LAMA2-EXOs showed faster recovery than did the control group. Although EXOs had a protective effect on endothelial cells, there was no LAMA2-enhanced protection on these cells. This is the first report of LAMA2-EXOs used to treat cardiomyocytes that underwent ischemia-reoxygenation injury. Overall, we showed that membrane-specific EXOs may help improve cardiomyocyte survival in treating ischemic cardiovascular disease.

## 1. Introduction

Ischemic heart disease predominantly presents as myocardial infarction (MI) or ischemic cardiomyopathy and is a significant cause of global morbidity and mortality [1] with profound implications for both clinical management and healthcare costs [1]. MI, a direct consequence of sustained oxygen deprivation, causes cardiac ischemia and results in significant cardiomyocyte loss. Immediate reperfusion, an established therapeutic method used to attenuate infarct size, paradoxically can exacerbate myocardial injury [2]. The limited regenerative capacity of adult mammalian cardiomyocytes provides challenges to myocardial repair and impedes intrinsic myocardial rejuvenation [3].

In recent decades, investigators in cardiac regenerative medicine have tried to leverage cell-based therapies for potential repair or replacement of damaged cardiomyocytes [4]. Despite promising preclinical outcomes, translational applications in clinical trials have yielded inconsistent and occasionally suboptimal results, regardless of cell type studied [5,6]. Although some studies have shown marginal improvements in heart function after stem cell engraftment in patients with MI, evidence of cell integration or new cardiomyocyte generation is lacking [7]. Instead, these beneficial effects have been attributed to immunomodulatory mechanisms, intricate paracrine signaling cascades, and growth factor secretion from transplanted cells [8]. Among the broad spectrum of trophic factors secreted by mesenchymal stromal cells (MSCs), extracellular vesicles have been reported to be responsible for these protective responses [9].

One type of extracellular vesicle that is of particular importance is the exosome (EXO), which is a nanosized intercellular particle that helps regulate diverse cell and molecular activities, such as cell proliferation, extracellular matrix (ECM) dynamics, inflammatory responses, and immune homeostasis. These vesicles harbor and transport an array of bioactive molecules, such as nucleic acids and proteins, with remarkable target specificity and physiological efficiency [10]. Given the cardiovascular system’s intricate cell heterogeneity [11], there is a compelling need to engineer cardio-specific exosomes. 

Laminins, specifically the cardiac isoforms laminin 211 and 221, which contain the alpha 2 chain (LAMA2), are crucial for myocardial integrity. These isoforms of laminin play a vital role in the cardiac ECM, ensuring myocardial structural cohesion [12,13] and potentially enhancing targeted uptake by cardiomyocytes [14]. In this study, we focused on a biomimetic polymer of laminin, polylaminin (polyLN), which effectively mimics the in vivo configuration and activity of these specific laminins [15,16]. By using the unique properties of polyLN in the context of LAMA2, an integral constituent of the cardiac extracellular environment, we aim to guide the production of MSC-derived exosomes toward a specialized cardiac-specific profile. 

Our overall hypothesis is that these LAMA2-enriched exosomes (LAMA2-EXO), generated in the presence of polyLN mimicking the cardiac-specific laminins 211 and 221, would have an enhanced affinity toward human-induced pluripotent stem cell (hiPSC)-cardiomyocytes. This affinity is expected to amplify their recognition and subsequent uptake by these target cells. This optimized interaction may provide a heightened protective shield for hiPSC-cardiomyocytes exposed to the harmful effects of ischemia-reoxygenation injury. If validated, this novel therapeutic strategy could allow for the tailoring of EXOs to specific cell targets, thus significantly improving ischemic heart disease interventions.

## 2. Results

LAMA2-EXOs were derived from MSCs cultured on polyLN211 and polyLN221 (Figure 1). Cells grown on the polyLN substrates maintained a fibroblastic morphology, with an elongated shape conducive to adhesion, similar to control cells grown on gelatin (Figure 1A). Nanoparticle tracking analysis revealed that the extracellular vesicles from MSCs under all conditions produced exosomes of the expected size range [17], averaging 120 nm (polyLN211, 127.6 ± 3.45 nm; polyLN221, 121.4 ± 11.58 nm; gelatin, 121 ± 14.4 nm) (Figure 1B,E). Exosome production was quantified using two distinct methods: particle size within the 120 nm range and expression of the exosomal marker CD63. For size-based quantification, cells yielded EXO concentrations of 1.13 × 10^11^ ± 0.50 × 10^11^ for polyLN211 and 0.73 × 10^11^ ± 0.27 × 10^11^ for polyLN221 (Figure 1C). For CD63+ EXOs, the concentrations were 0.96 × 10^11^ ± 0.31 × 10^11^ for polyLN211 and 1.11 × 10^11^ ± 0.31 × 10^11^ for polyLN221 (Figure 1D). In the EXO-gelatin control group, similar concentrations were observed, with 1.17 × 10^11^ ± 0.34 × 10^11^ and 1.13 × 10^11^ ± 0.13 × 10^11^ based on particle size and 1.13 × 10^11^ ± 0.13 × 10^11^ based on CD63 expression, respectively. 

RNA sequencing was performed to verify the mRNA profile of the EXOs produced under LAMA2 coating. We identified 798 RNAs that were significantly (*p* ≤ 0.05) differentially expressed between the LAMA2 groups. When compared to gelatin, we detected 377 differentially expressed genes for polyLN211 and 731 genes for polyLN221, respectively (Figure 2). Unsupervised hierarchical clustering revealed a strong correlation within groups (Figure 3A), with gene expression linked to cell metabolism, structure, mitochondrial activity, and cell–ECM interactions (Appendix A). When the *p* value was adjusted to 0.01, 42 RNAs were significantly differentially expressed in LAMA2 groups: 10 comparing polyLN211 and gelatin and 32 comparing polyLN221 and gelatin (Figure 3B). Note that one of the genes with a higher differential expression in polyLN211-EXOs is the 70 kDa heat shock protein (*HSP70*), which is important in controlling cell proteostasis [18]. Thymosin beta 4 X-link (*TMSB4X*) and thymosin beta 10 (*TMSB10*) were increased in polyLN221-EXO in comparison with gelatin-EXOs; these genes regulate actin polymerization, cell migration, proliferation and differentiation, and calmodulin 1, which modulates cardiac ion channels (Figure 3B and Appendix A) [19,20,21]. PolyLN211- and polyLN221-EXOs presented 20 RNAs that were differentially expressed. Most of these RNAs were involved with ribosomal activity, the ECM, cell contractility, endothelial differentiation factors, cardiomyocyte hypertrophy, and increased levels of cell–cell and cell–matrix interactions (Appendix A). 

The RNA-sequencing data showed that 9 of 42 differentially expressed mRNA were reported for the first time on EXOs (Figure 3B and Appendix A). These data were validated by Vesiclepedia, a web-based compendium of RNA detected or associated with extracellular vesicles [22]. We found differential expression of cardiac ncRNA NEX antisense RNA 1 (*NEXN_AS1*) in polyLN211-EXOs versus gelatin. We also first observed the presence of microtubule crosslinking factor (*MTCL1*), asialoglycoprotein receptor 2 (*ASGR2*), and cell division cycle 27 pseudogene 9 (*CDC27P9*) in polyLN221 and gelatin, and mRNAs related to carbohydrate metabolic processes, such as LanC Like Family Member 3 (*LANCL3*) and EvC ciliary complex subunit 2 (*EVC2*), were decreased in LAMA2_EXO compared to gelatin. Although the Indian hedgehog signaling molecule (IHH), the RNA component of signal recognition particle (*RN7SL2*), and the FAT Atypical Cadherin (*FAT3*) are listed as proteins in Vesiclepedia, we report them as mRNA. 

SH3 domain-binding glutamate-rich protein like 3 (*SH3BGRL3*), a gene involved in cytoskeletal organization, was uniformly upregulated across all groups exhibiting a higher expression in polyLN221-EXO compared to gelatin (log fold change = 9.06). Specifically, polyLN211-EXO exhibited a log fold change of 7.55 higher when compared to polyLN221-EXO (Appendix A). This gene’s interaction with unconventional myosin 1c suggests a meaningful role in cell motility, mRNA maturation, and chromatin remodeling [23]. Functional enrichment analysis (STRING) predicted significant protein–protein interactions for SH3BGRL3, including epidermal growth factor receptor [24], which is pivotal for translating ECM signals into specific cellular responses (Figure 3C).

To observe the effect of EXO-LAMA2 on endothelial cells (ECs) and cardiomyocytes under normal culture conditions, we evaluated EXO uptake and observed the internalization of EXO into the cell cytoplasm after 2 h (Figure 4A). EC function, evaluated by tube formation, was not impaired by the presence of EXOs (Figure 4B). Cell counts were measured in ECs and cardiomyocytes at baseline (day 0) and on days 1–3 (EXO treatment on day 1). Our findings showed no difference in the number of ECs and cardiomyocytes over time between treated and control cells (Figure 4C). Furthermore, EXO treatment did not have a cytotoxic effect on the cells over time (Figure 4D). 

To establish a robust in vitro model that closely mimics ischemia-reperfusion injury as seen in vivo, we conducted preliminary studies of cell growth in varying O_2_ concentrations and media compositions. We used an ischemic medium (IRI media), characterized by low glucose levels and a reduced pH (6.4), to simulate ischemic nutritional conditions. The results of these preliminary studies (Appendix A) were instrumental in determining the most effective experimental design for our ischemic-reoxygenation model. In the primary experimental model, we cultured ECs and cardiomyocytes under hypoxic or normoxic conditions with either ischemic or control media for 4 h and then evaluated cell morphology and membrane integrity at 4 h post-ischemia and at 1.5 h and 3 h post-reoxygenation (Appendix A). Our control groups for these experiments were cells maintained in normoxic conditions with control media. Additionally, we included groups combining normoxia with ischemic medium and hypoxia with control medium to provide a comprehensive analysis of the different environmental impacts on cell health. 

For ECs cultured under normoxic conditions with control media, cells exhibited a flat, polygonal morphology at all timepoints (Appendix A); however, after 4 h in IRI media, floating dead and shrunken cells were seen immediately post-ischemia (Appendix A). A significant cell recovery was observed 1.5 h post-reoxygenation (Appendix A), and by 3 h post-reoxygenation, ECs exhibited a morphology similar to that of the control group (Appendix A). For ECs under hypoxic conditions (Appendix A), we noted a higher toxicity with more detached and shrunken cells after 4 h of hypoxia (Appendix A). Minimal recovery was observed at 1.5 h and 3 h of reoxygenation (Appendix A), particularly in cells subjected to both hypoxia and IRI media (Appendix A). Similar effects were observed for cardiomyocytes under these conditions (Appendix A). 

To validate membrane integrity, we conducted a cytotoxic lactate dehydrogenase (LDH) assay; EC/cardiomyocyte media under normoxia served as our control condition. Under normoxic conditions, no significant difference in cytotoxicity was noted between IRI and control media (EC or cardiomyocyte media) over time in either ECs or cardiomyocytes (Appendix A). In contrast, substantial cytotoxicity was observed after 4 h of hypoxia with IRI media compared to control media. For ECs, after hypoxia, the cells in IRI media showed significantly higher cytotoxicity compared to control media under both normoxia and hypoxia (hypoxia with IRI media 1.91 ± 0.42-fold, hypoxia with control media 1.14 ± 0.36-fold, and normoxia with control media 1.00 ± 0.43). For cardiomyocytes, high cytotoxicity (>95%) was observed in IRI media under hypoxic conditions (Appendix A). These results, combined with the less severe morphological injury in the presence of control media, suggest that the combination of IRI media and hypoxia is necessary to produce substantial cell injury after 4 h. 

After establishing that the combination of hypoxia with IRI media for 4 h followed by reoxygenation was the most effective model for simulating ischemia-reperfusion injury in vitro, we examined the protective effects of different EXOs on ECs and cardiomyocytes (Figure 5A). We pre-treated cells with EXO-gelatin, EXO-polyLN211, and EXO-polyLN221 to determine their potential in safeguarding the cells against the detrimental effects of our ischemia-reoxygenation model (Figure 5B–E). After 4 h of hypoxia, ECs showed fewer morphological changes (e.g., cell shrinkage and enlargement indicated by orange arrows in Figure 5F–I) and were able to recover after 3 h of reoxygenation (Figure 5O–Q); this differed from the control group, which showed an increased number of injured cells at all timepoints (Figure 5F,J,N, orange arrows). This effect was also observed in cardiomyocytes with fewer dead cells post-reoxygenation (Figure 6) compared to ECs.

After pre-treating cells with EXO for 2 h, we noted the protective effects of EXOs on ECs and cardiomyocytes post-hypoxia (Figure 7). For ECs, all three EXOs protected the cells from hypoxia (4 h hypoxia group) and reoxygenation (time 1.5 h and 3 h), as the LDH values were similar to baseline values (Figure 7A). In contrast, hypoxia caused higher cytotoxicity in cardiomyocytes, and EXO-polyLN221 was more protective compared to non-treated cells (Figure 7B). After 1.5 h of reoxygenation, we observed an enhanced recovery of membrane integrity in LAMA2-EXO cells compared to those without treatment (Figure 7B). After 3 h post-ischemia, all experimental groups recovered to baseline values. Analogous to membrane integrity, the EXOs’ protective effects were observed by measuring the cellular apoptosis activation pathway (caspase 3/7) (Figure 7B,C). The cells reached maximum caspase 3/7 activity values post-hypoxia (4 h hypoxia groups), after which EXO treatment improved cell recovery following reoxygenation (time 1.5 h and 3 h) compared to the control group. Although there was no statistical difference between gelatin-EXOs and LAMA2-EXOs after 1.5 h of reoxygenation, we observed a trend toward a faster recovery of cardiomyocytes in the presence of LAMA2-EXOs (Figure 7). 

## 3. Discussion

One persistent challenge in the application of MSCs for cardiovascular therapy has been the inherent complexity of the target organ and the typically poor engraftment of transplanted cells at the injury site [25,26]. EXOs, as critical mediators of intercellular communication, have a composition that is highly reflective of their cellular origin and the dynamic interplay within the surrounding environment and intracellular pathways [27,28]. Because of their capacity to convey bioactive molecules across organ systems, EXOs have emerged as a focal point in cardiac repair strategies and may have the potential to replace cell therapies. Studies of the functional heterogenicity of EXOs have shown that even small variations in their content can result in a range of recipient cell outcomes, from enhanced survival and immunomodulation to apoptosis [29]. Advancements in next-generation sequencing have improved our understanding of exosomal cargo and thus their therapeutic use [30]. Here, we have demonstrated that closely mimicking the plasma membrane epitopes of host cells by using LAMA2-EXOs facilitates targeted delivery and enhanced EXO uptake by cardiomyocytes, leading to a noticeable protective effect against ischemia-reoxygenation injury. We have also demonstrated that altering the ECM to cultivate MSCs was sufficient to produce EXOs enriched in RNAs associated with critical cellular functions and cytoskeletal dynamics. Additionally, this RNA cargo has the potential to activate signaling pathways that can promote myocyte survival under ischemic conditions while preserving the immunomodulatory essence of EXO-MSCs. By harnessing the unique capabilities of these exosomes, we envision a new paradigm in the treatment of ischemic heart diseases in which targeted delivery and specificity of action could significantly enhance therapeutic outcomes.

The cardiac ECM consists of an intricate framework of proteins, including collagen, elastin, fibronectin, glycosaminoglycan, proteoglycan, and laminin. Each contributes to the myocardium’s structural and functional integrity. Beyond providing mechanical support, the cardiac ECM is dynamically interactive and orchestrates a range of signaling cues essential for cell survival and specialized cardiac functions [31]. Cardiac cells modulate the ECM via a continuous remodeling process and thus dictate key aspects of cell behavior such as survival, proliferation, migration, and differentiation [32]. Integral to this matrix are laminins, a family of heterotrimeric glycoproteins composed of α, β, and γ subunits. Laminins contribute to the cardiac scaffold while serving as crucial chemical cues that elicit a spectrum of cellular responses essential for cardiac function and response to injury. The α subunit (chains) has cell attachment sites that bind to the plasma membrane via integrins, facilitating critical cell–ECM interactions [13,33,34].

The α2 chain is notably linked to the specialized functions within striated muscles and peripheral nerves, underscoring its specialized role in these tissues [35,36]. Here, we opted to use the α2 laminin chain to increase the homing of these EXOs to cardiomyocytes because the laminin α2 chain is linked to striated muscle functions. hiPSC-derived cardiomyocytes have an immature phenotype relative to adult cells [37]. We had previously shown that the ECM can affect cardiac maturation by promoting the transition from immature to mature laminin ligand (integrin) phenotypes [38]. Acknowledging the transition of β1 to β2 laminin chains during cardiac maturation [39], we strategically chose to utilize both 211 and 221 laminins for a comparative analysis of their effects. The ECM composition of ECs is mainly formed by laminins 411 and 511, both of which are vital in vascular homeostasis [40,41]. This distinction in the ECM composition between ECs and cardiomyocytes may have contributed to our finding that LAMA2-EXOs showed enhanced cardiomyocyte protection compared to their effects on ECs. In addition, this finding emphasizes the tailored effectiveness of LAMA2-EXO in cardiac applications, leveraging specific laminin interactions to support cardiomyocyte resilience under ischemic conditions. 

Ischemic injury triggers a spectrum of cellular responses, including processes that cause swelling and/or bursting of the cell and thus result in damage to the plasma membrane [42]. In addition, the activation of signaling cascades that lead to cell death is closely related to ischemic injury and cell remodeling after injury [43]. A common method used to examine the activity of cytoplasmic enzymes released by damaged cells is the LDH assay because this enzyme is rapidly released into the culture supernatant when the plasma membrane is damaged [44]. Apoptosis is considered a vital component of chemical-induced cell death and ischemic damage, and reports indicate that this process can be reversed if the apoptotic stimulus is removed at an early stage [45]. Here, we assessed cell damage by measuring both LDH and caspase 3/7 and examined cell morphology to determine the protective effects of EXOs on cardiomyocytes and ECs under ischemic injury. Studies have shown the protective effect of extracellular vesicles from different origins on cardiomyocytes with hypoxic injury [46,47]. In one study, the levels of LDH were approximately two-fold higher in cardiomyocytes with hypoxia than in those with no injury, even with pretreatment with extracellular vesicles; however, treatment with extracellular vesicles protected cardiomyocytes from injury as evidenced by lower LDH increases versus controls when compared to the increase seen in non-treated cells [48]. Although we observed similar results in ECs under all EXO treatments, our data demonstrate an enhanced protective effect of LAMA2-EXOs in cardiomyocytes, mainly after 1.5 h of reoxygenation compared to untreated cells; LAMA2-EXO treatment resulted in cells reaching baseline values more quickly than those treated with gel-EXOs. EXOs have anti-apoptotic characteristics, and this protection has been previously studied [49]. Similar to our findings, a recent study demonstrated a protective effect of EXOs, with a 60% decrease in apoptosis in cardiomyocytes under ischemia-reperfusion injury [50]. Our in vitro data could be used to further improve results in the acute MI animal models where earlier EXO treatment in combination with late MSC delivery slightly enhanced cardiac function [51]. Targeting cardiomyocytes with LAMA2-EXOs may boost cardiomyocyte survival and reduce infarction progression, while maintaining the pan-protective effect of EXOs.

Although further study is needed to fully decipher the mechanisms governing EXO uptake and their subsequent therapeutic effects, our RNA sequencing analysis has revealed a unique mRNA profile in EXOs produced under the influence of LAMA2. This aligns with previous studies that have underscored the complex interplay between EXOs and the ECM in which EXOs contributed to the physical and bioactive structural components of the ECM [52] and modulated ECM composition by activating specific cell pathways [53]. Here, we demonstrated that MSCs from the same patient, when cultivated on different substrates (gelatin vs. LAMA2), exhibit distinctive EXO cargo. This observation differs from previous studies, in which the EXO influenced the ECM composition, reinforcing the essential role of the ECM on EXO biogenesis. Specifically, LAMA2 substrate fostered the production of EXOs enriched with genes associated with critical cellular functions such as motility, metabolism, intracellular trafficking, and cytoskeletal dynamics, including actin production and activity, cell contraction, and cell junction assembly and organization. Additionally, this RNA cargo has the potential to activate signaling pathways that can promote myocyte survival under ischemic conditions while preserving the immunomodulatory essence of EXO-MSCs. Together, these insights affirm the pivotal role of the ECM in EXO biogenesis and highlight the therapeutic promise of using specific ECM components to customize exosomal cargo for precise regenerative treatments.

This study, while providing promising insight into the therapeutic potential of LAMA2-EXOs, has limitations. Initially, we used MSCs sourced from three heart-failure patients in the FOCUS-CCTRN trial who showed improved clinical function. However, these samples represent a small, selective cohort. Regrettably, the destruction of remaining samples from other patients not included in this ancillary study limits the breadth of our analysis and the availability of material for further study. To build on the foundational knowledge we have established, extensive research is essential. Future endeavors should focus on replicating and critically examining our findings across additional models of cardiovascular ischemia/reperfusion injury. This would facilitate a more comprehensive understanding and validation of the therapeutic potential and the mechanisms of LAMA2-EXOs across a broader patient spectrum. Such efforts are vital for translating our initial in vitro insights into viable, effective therapeutic strategies for heart failure, ultimately paving the way for significant advancements in patient care and treatment outcomes.

## 4. Materials and Methods

### 4.1. MSC Expansion

MSCs (*n* = 3) were isolated from bone marrow mononuclear cells from patients in the clinical improvement group of the FOCUS-CCTRN trial [54]. The cells were grown in Dulbecco’s Modified Eagle Medium (DMEM) (1×)+GlutaMAX-I (Gibco, Carlsbad, CA, USA) supplemented with 10% of characterized fetal bovine serum of U.S. origin (HyClone, South Logan, UT, USA) and 1% of penicillin–streptomycin (Gibco). We expanded the cells under standard culture conditions (37 °C at 5% CO_2_) for over 4 passages and used approximately 2 × 10^6^ cells to produce the EXOs. Bright-field images were acquired using a high-performance color CMOS C-mount microscope camera and AmScope 3.1 software (AmScope, Irvine, CA, USA).

### 4.2. Exosome Production and Concentration

For MSC-EXO production, we coated plates with polyLNs—the biomimetic form of human recombinant LN211 or LN221 (100 µg/mL, BioLamina, Sundbyberg, Sweden), (Appendix A). We diluted the polyLNs in cold, sterile acid buffer solution (20 mM sodium acetate, 1 mM CaCl_2_, pH 4.0) at final concentrations of 5 µg/mL (polyLN211 and polyLN221) as previously reported [55]. Then, 10 mL of polyLN211 or polyLN221 was added to a 175 mm^2^ flask, which was maintained overnight at 37 °C. The next day, we washed the flasks 3 times with phosphate-buffered saline (PBS) before plating the cells. Gelatin at a final concentration of 0.5% was used as a coating control. MSCs were cultured under different coating conditions for 2 days (Appendix A), and on day 3, the medium was changed to serum-free DMEM (1×)+GlutaMAX-I (Gibco) for EXO production. 

After 24 h, the supernatant was collected (Appendix A) and centrifuged at 2000× *g* for 30 min at 4 °C for debris removal. Both the polyLN-EXOs and gelatin-EXOs were concentrated using an Invitrogen (Waltham, MA, USA) Total Exosome Isolation Reagent (from cell culture media). Briefly, 25 mL of the supernatant was diluted in a 2:1 ratio with the Total Exosome Isolation reagent and incubated overnight at 4 °C. Then, the samples were centrifuged at 10,000× *g* for 1 h at 4 °C, and the EXO pellet was resuspended in 1 mL of PBS and stored at −80 °C until further use (Appendix A).

### 4.3. EXO Quantification and Characterization

The amount of MSC-EXO produced was quantified by measuring the expression of CD63 antigen and by performing a nanoparticle tracking analysis. To achieve this, we used an EXO-ELISA kit (System BioSciences Cat #EXEL-ULTRA-CD63-1, Palo Alto, CA, USA), which is a sensitive enzyme-driven technology, for quantifying the number of EXOs present in the samples. Briefly, EXOs were dosed on a protein-coated plate and combined with a CD63 antibody, which recognized the presence of the exosome surface protein. A peroxidase secondary enzyme was then used for signal detection and amplification. The assay plate was read with a colorimetric substrate additive. Absorbance was measured using a plate reader at 450 nm (Tecan, Zürich, Switzerland). Nanoparticle tracking analysis and transmission electron microscopy were performed and analyzed by Alpha Nano Tech LLC (Research Triangle Park, NC, USA).

### 4.4. EXO mRNA Sequencing

RNA was extracted from exosomes derived from cell culture supernatant using the exoRNeasy Midi Kit (Qiagen, Venlo, The Netherlands) following the manufacturer’s instructions. Extracted RNA was quantified using a Qubit Fluorometer High Sensitivity RNA assay (Thermo Fisher Scientific, Eugene, OR, USA). The RNA libraries were prepared and sequenced at the University of Houston’s Seq-N-Edit Core laboratory per standard protocols. RNA was enriched, and subsequent mRNA libraries were generated with a QIAseq^®^ Stranded mRNA Library Kit. The size selection for libraries was performed by using SPRIselect beads (Beckman Coulter, Indianapolis, IN, USA), and the library purity was analyzed with High Sensitivity d1000 ScreenTape using Agilent Tapestation 4200. The libraries were pooled and sequenced using Illumina NovaSeq 6000, generating ~30 million 2 × 150 bp paired end reads per sample. Raw sequencing data (Fastq) were loaded into the CLC Genomics Workbench (Version 20.0.4 Qiagen) for analysis. The Illumina sequencing adaptors were trimmed, and reads were mapped to the human reference genome hg38 Refseq GRCh38.p9. Read alignment was represented as integer counts by using parameters of mismatch cost 2, insertion cost 3, deletion cost 3, length fraction 0.8, similarity fraction 0.8, with a maximum of 10 hits per read. Integer read counts were normalized by the Trimmed Means of M-values (TMM) algorithm. The FDR-adjusted *p*-value cut-off was set at 0.05, and gene expression differences were calculated based on log-fold changes of 0.5. The list of differentially expressed mRNA (*p* < 0.01) was validated using Vesiclepedia’s data 2024. 

### 4.5. Cardiomyocyte Differentiation

To obtain human cardiac cells, we differentiated hiPSCs (SCVI20) toward cardiac lineage by using a commercially available differentiation kit (STEMCELL Technologies Inc., Cambridge, MA, USA). Briefly, 1.2 × 10^6^ hiPSCs were plated on 6-well tissue culture plates pretreated with hESC-qualified Matrigel (Corning, Bedford, MA, USA) in TeSR1 E8 medium (STEMCELL), supplemented with ROCK Inhibitor Y-27632 (ATCC, Manassas, VA, USA). The supernatant was aspirated the next day and replaced with fresh TeSR1 E8 medium. Cardiac differentiation began on day 0, with the addition of STEMdiff Cardiomyocyte Differentiation Medium A supplemented with Matrigel Matrix Basement Membrane, Growth Factor Reduced (Corning). The medium was changed to STEMdiff Cardiomyocyte Differentiation Medium B on day 2, and STEMdiff Cardiomyocyte Differentiation Medium C on days 4 and 6. Cells were maintained in STEMdiff Cardiomyocyte Maintenance Medium from days 8 to 15, with media replacement every other day. Cardiomyocytes were harvested using the STEMdiff Cardiomyocyte Dissociation Kit (STEMCELL) for use in experiments.

### 4.6. Ischemia Injury Model

To create the ischemia injury model (Appendix A), we tested the following combination of groups: (1.1) cells under hypoxia (1% O_2_ and 5% CO_2_ at 37 °C) or (1.2) normoxia (20% O_2_ and 5% CO_2_ at 37 °C) cultured for 4 h in the presence of (2.1) ischemic medium (IRI medium, composed of DMEM without glucose [Gibco] and an adjusted pH of 6.4) or (2.2) control medium composed of Endothelial Cell Growth Medium-2 (Lonza, Cambridge, MA, USA) for ECs, and STEMdiff Cardiomyocyte Maintenance Medium for cardiomyocytes. Cells in reoxygenation (20% O_2_ and 5% CO_2_ at 37 °C) were cultured with control medium. Cells cultured under hypoxia and IRI medium were considered the experimental groups, and cells cultured under normoxia and EC or cardiomyocyte media were considered the control group.

To simulate ischemia injury in vitro, we plated 1 × 10^4^ human umbilical vein endothelial cells from pooled donors (Lonza) and 1 × 10^5^ hiPSC-CMs on 96-well tissue culture plates pretreated with gelatin and Matrigel (Corning), respectively, 2 days before exposure to 4 h of hypoxia. Ischemic medium was used to initiate cell injury during the 4 h hypoxia period. Then, the cells were removed from the chamber, and the culture media were aspirated and replaced with EC and cardiomyocyte media, respectively. The plates were transferred to normoxic culture conditions (5% CO_2_, 21% O_2_ at 37 °C) for 1.5 h and 3 h of reoxygenation (Appendix A). To test the protective effect of EXOs, we incubated the cells with 10^9^ particles of gelatin-EXOs, polyLN211-EXOs, or polyLN221-EXOs 2 h before the initiation of ischemia (Figure 4A, Appendix A).

### 4.7. Cell Damage

A homogeneous membrane integrity assay (Promega, Madison, WI, USA) was performed to determine the amount of LDH released into the culture medium by cells with damaged membranes. First, control and EXO-treated cardiomyocytes and ECs were cultured in 96-well plates and exposed to normoxic or hypoxic conditions. Lysis solution was added to serve as a positive control. At each experimental timepoint (baseline, 4 h post-ischemia/4 h post-normoxia, 1.5 h of reoxygenation, and 3 h of reoxygenation), the plates were equilibrated to room temperature, and 100 µL of the CytoTox-ONE reagent was added to each well for a 10-min incubation. Afterwards, we added 50 µL of stop solution and measured the fluorescent signal by using a plate reader with an excitation wavelength of 560 nm and an emission wavelength of 590 nm (Tecan). The toxicity was calculated by using the average fluorescence values from experimental, maximum LDH release (positive control), and culture medium background wells.

A homogeneous luminescent assay was performed to measure caspase-3 and -7 activities (Promega). After normoxic or hypoxic exposure and reoxygenation timepoints, the plates were equilibrated to room temperature, and 100 µL of the Caspase-Glo^®^ 3/7 reagent was added to each well for a 45 min incubation. Blank and negative controls (normoxia, no EXO treatment) were used to determine background and base values. The luminescence was measured by using a plate-reading luminometer (Tecan).

### 4.8. Cell Viability and Proliferation

We used the Cell Counting Kit-8 (cck8, Sigma-Aldrich, Steinheim am Albuch, Germany) to evaluate cytotoxicity, cell attachment, and proliferation of ECs and cardiomyocytes after EXO treatment on days 0 (baseline), 1, 2, and 3. We tested the following experimental condition: 1 × 10^4^ of ECs and 1 × 10^5^ hiPSC-CMs plated on 96-well tissue culture plates and EXOs were added to the culture medium 24 h later. Cells were incubated with 0.01 μL mL^−1^ of CCK-8 solution under standard culture conditions (37 °C at 5% CO_2_) for 4 h. Cell supernatant was collected, and the absorbance was measured at 450 nm using a microplate reader (Tecan).

### 4.9. Tube Formation Assay

A 96-well plate was pretreated with 50 µL of Matrigel Matrix Basement Membrane, Growth Factor Reduced for 2 h at 37 °C, and 2 × 10^4^ ECs were seeded in the presence or absence (control) of LAMA2-EXOs or gelatin-EXOs under standard culture conditions. After 16 h, brightfield images were captured to verify qualitatively the production of tube-like structures.

### 4.10. Exosome Uptake

EXOs were labeled with a ExoGlow-Membrane EV Labeling Kit (Cat #EXOGM600A-1, System Biosciences, Palo Alto, CA, USA) in which a labeling reaction buffer was prepared in a 1:6 dilution of labeling dye. Next, 14 µL of the reaction buffer was added to the EXOs, which were then incubated for 30 min at room temperature. To remove excess dye, we isolated the labeled EXOs by using the Invitrogen Total Exosome Isolation Reagent as described above. We plated 0.5 × 10^4^ of ECs and cardiomyocytes, and after 24 h of recovery, the cells were cultured in the presence of the labeled EXOs under standard conditions. The EVOS M700 was used to capture the internalization of gelatin-EXOs and LAMA2-EXOs into the cells over time.

### 4.11. Statistical Analysis

Data are shown as the mean ± standard deviation. Samples were compared by using a Student’s *t* test or two-way analysis of variance, and *p* < 0.05 and *p* < 0.01 were considered statistically significant. GraphPad Prism software, version 9.3.1 (GraphPad Software Inc., La Jolla, CA, USA), was used for statistical analyses. 

Differential gene expression analysis was performed using CLC Genomics workbench (Version 20.0.4 Qiagen). The data were normalized by the TMM algorithm. The expression level was modeled by generalized linear model, and the Benjamini–Hochberg correction method was used as a false discovery rate adjustment for multiple tests.

## 5. Conclusions

This study has advanced stem cell research by demonstrating that EXOs with cardiac-specific ECM properties can be tailored to enhance cardiomyocyte survival without diminishing benefits to other cell types. Using next generation sequencing, we have described the unique molecular composition of these EXOs and have shown how the ECM can affect their content and function. In addition, our study is the first to quantify the mRNA content under different ECM substrates. Our approach introduces a targeted therapeutic strategy, suggesting that modifying the microenvironment can refine EXOs to specifically support heart tissue repair. These findings open new possibilities for EXO-based therapies in treating cardiovascular diseases and emphasize precision over generalized treatment methods.

## Figures and Tables

**Figure 1 ijms-25-03773-f001:**
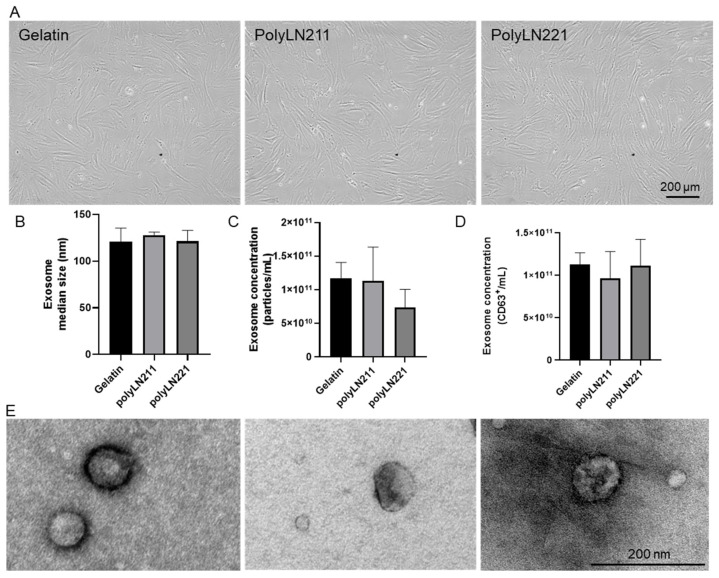
Exosome production and characterization. (**A**) Brightfield images of human mesenchymal stromal cells (MSCs) cultured with gelatin, polyLN211, and polyLN221 substrates. (**B**–**E**) Exosome (EXO) characterization. (**B**) Size of EXOs (*n* = 3), (**C**) number of particles (*n* = 3), and (**D**) number of CD63+ particles generated using different substrates (*n* = 3). (**E**) Transmission electron micrographs demonstrating the EXO size generated under gelatin, polyLN211, and polyLN221, respectively.

**Figure 2 ijms-25-03773-f002:**
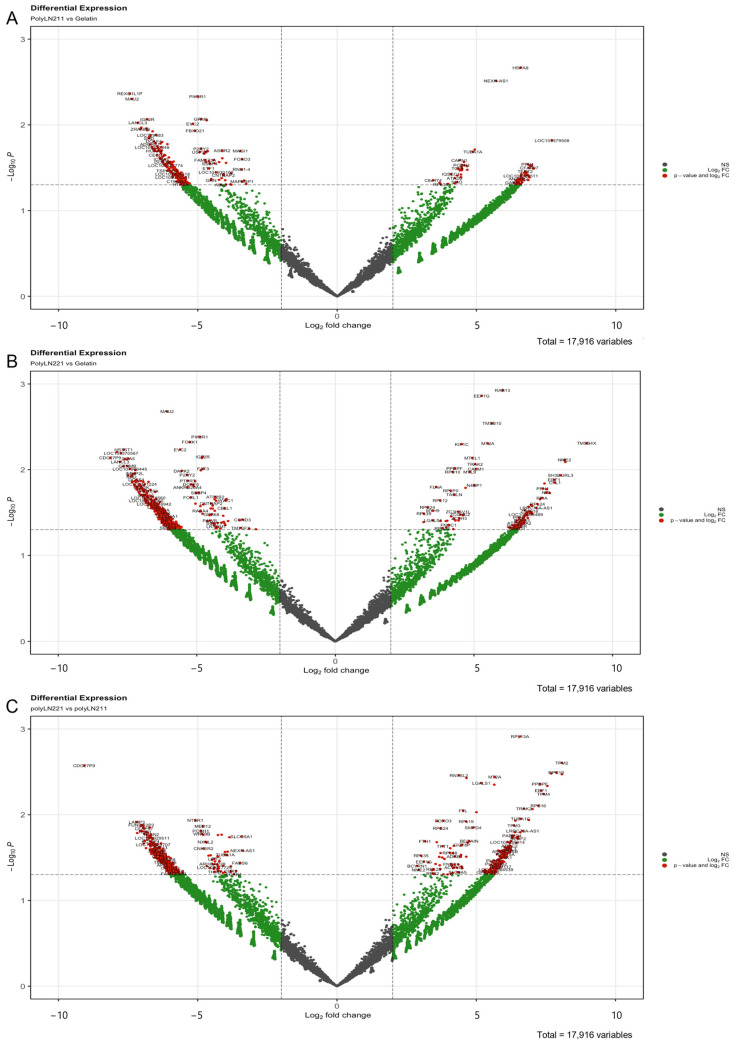
RNA-sequencing analysis of exosomes cultured under polyLN211, polyLN221, and gelatin. Volcano plot of EXOs under polyLN211 vs. gelatin (**A**), polyLN221 vs. gelatin (**B**), and polyLN211 vs. polyLN221 (**C**) (red, *p* ≤ 0.05).

**Figure 3 ijms-25-03773-f003:**
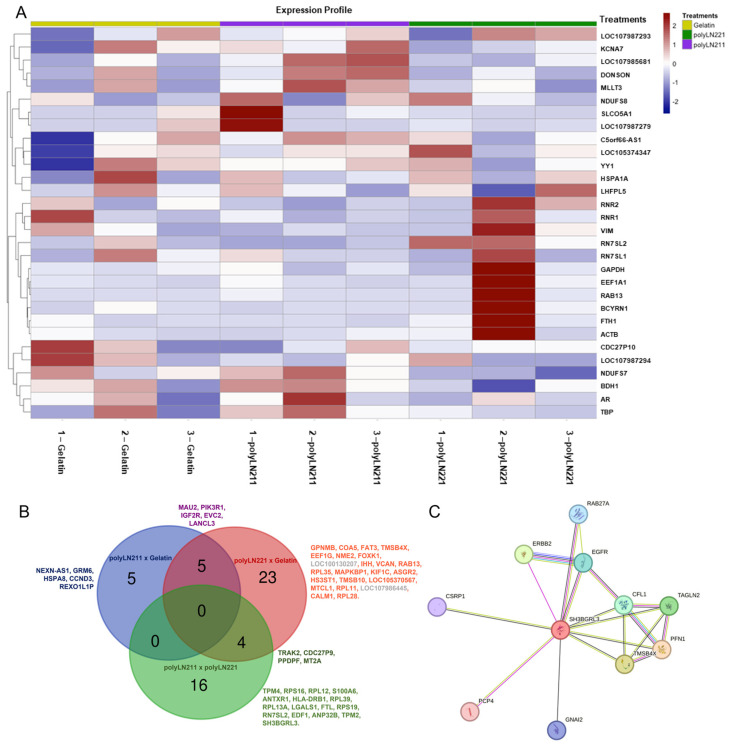
RNA-sequencing analysis of exosomes cultured under polyLN211, polyLN221, and gelatin. (**A**) Heat map of differentially expressed genes. (**B**) Venn diagram (*p* ≤ 0.01) of polyLN211 vs. gelatin (blue), polyLN221 vs. gelatin (red), and polyLN211 vs. polyLN221 (green). (**C**) Functional enrichment analysis of SH3 domain-binding glutamate-rich protein like 3 (*SH3BGRL3*) gene.

**Figure 4 ijms-25-03773-f004:**
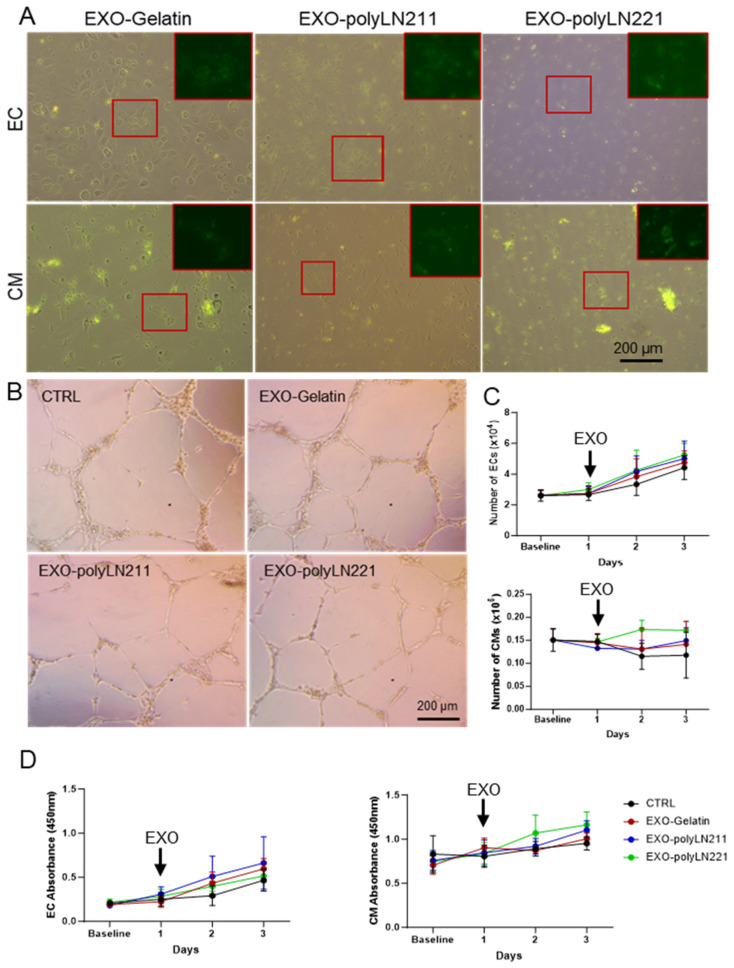
Effect of LAMA2-EXOs on endothelial cells (ECs) and cardiomyocytes (CM) under normal conditions. (**A**) EXO internalization after 2 h of incubation. Green indicates EXOs. Zoomed images of the red boxed areas are shown on the top right-side panel. (**B**) Tube formation assay after EC pretreatment with EXOs. (**C**) Number of ECs (top) and CMs (bottom) over time (*n* = 3). (**D**) Cytotoxicity and proliferation rate after EXO treatment on ECs (left) and CMs (right) over time (*n* = 3). The cells were treated with EXOs 24 h after plating, as indicated by the arrow.

**Figure 5 ijms-25-03773-f005:**
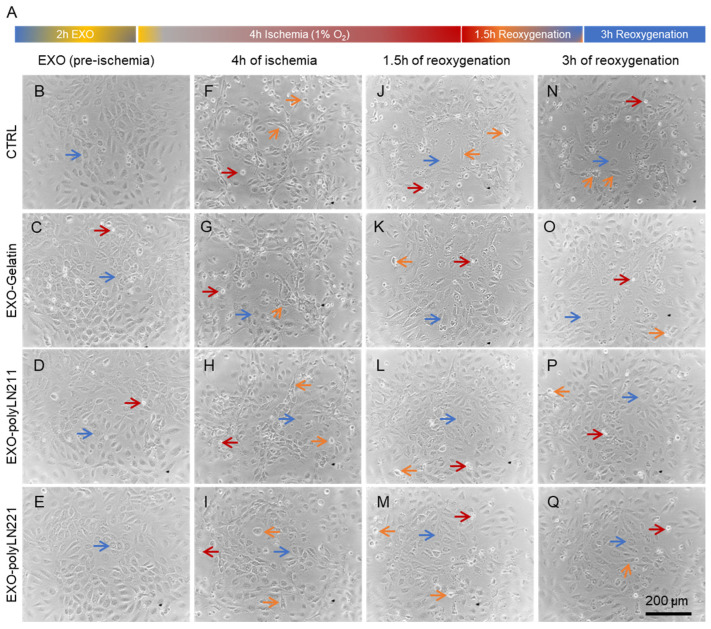
Protective effect of EXOs on endothelial cells in an ischemia-reoxygenation model. (**A**) Schematic diagram showing the main steps of ischemia reoxygenation in endothelial cells. (**B**–**Q**). Brightfield images of cells treated with different EXOs at pre-ischemia (**B**–**E**), 4 h after ischemia (**F**–**I**), 1.5 h of reoxygenation (**J**–**M**), and 3 h of reoxygenation (**N**–**Q**). Blue arrows, healthy cells; orange arrows, injured cells; and red arrows, (floating) dead cells.

**Figure 6 ijms-25-03773-f006:**
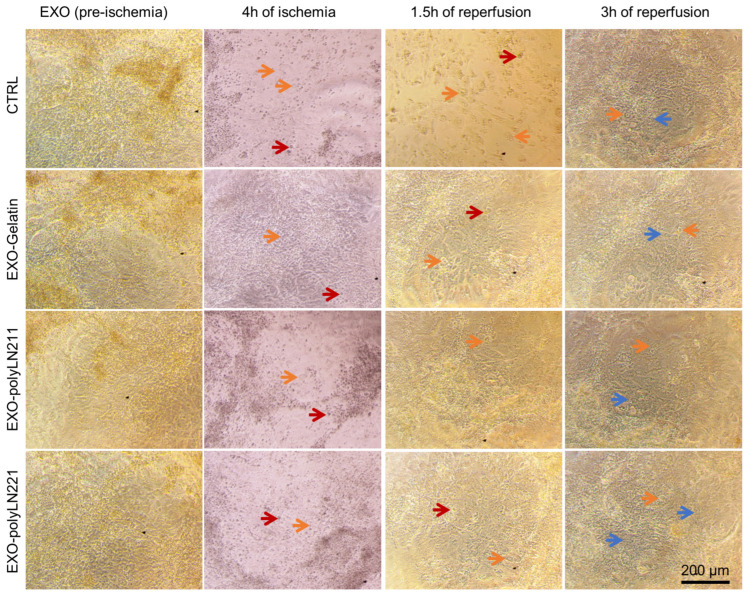
Protective effect of EXOs on cardiomyocytes in an ischemia-reoxygenation model. Brightfield images of cells treated with different EXOs. Blue arrows, healthy cells; orange arrows, injured cells; and red arrows (floating) dead cells.

**Figure 7 ijms-25-03773-f007:**
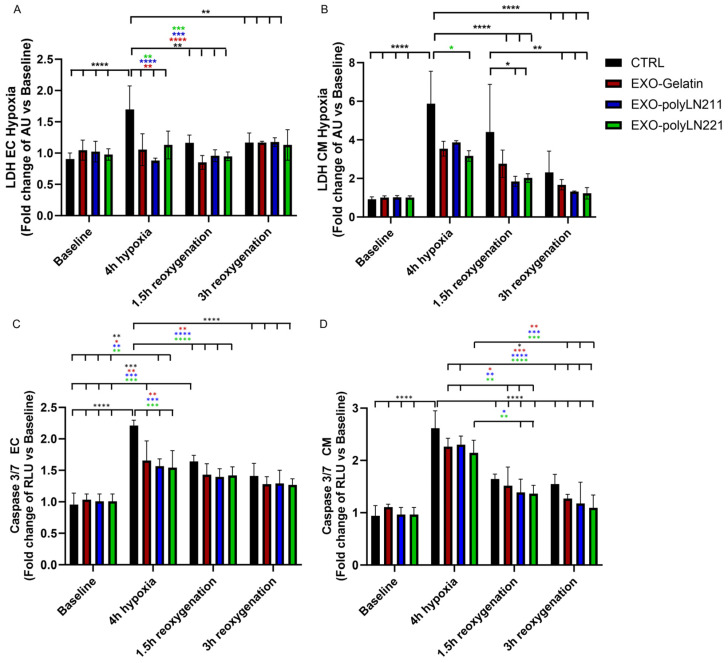
Cytotoxicity effect on endothelial cells (ECs) and cardiomyocytes (CMs) in an ischemia-reoxygenation model. Membrane integrity assay (LDH) and apoptosis activation (caspase 3/7) indicate the protective effect of EXO on ECs (**A**,**C**) and CMs (**B**,**D**). AU, arbitrary unit; CTRL, control; RLU, relative luminescence unit. Black * vs. CTRL, red * vs. EXO-Gelatin, blue * vs. EXO-polyLN211, green * vs. EXO-polyLN221. * *p* < 0.05; ** *p* < 0.01; *** *p* < 0.001; **** *p* < 0.0001.

## Data Availability

The data presented in this study are available on request from the corresponding author.

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
