# Peer review of "Laminin Alpha 2 Enhances the Protective Effect of Exosomes on Human iPSC-Derived Cardiomyocytes in an In Vitro Ischemia-Reoxygenation Model"

_ijms, 2024, doi:10.3390/ijms25073773_

Round 1

Reviewer 1 Report

Comments and Suggestions for Authors

The current study investigates the potential cardioprotective role of exosomes from mesenchymal stromal cells (MSCs) cultured under different conditions on both endothelial cells and iPSC-derived cardiomyocytes against ischemia and reoxygenation injury. Exploring extracellular vesicles with specific recognition of cardiac cells for therapeutic purposes is of great interest and may represent an innovative and more effective approach to alleviate cardiovascular diseases. While the manuscript is intriguing and generally well written, there are some important comments and issues that require further attention.

- The RNAseq experiment primarily characterizes the content of exosomes obtained from MSCs cultured with different coatings, but lacks validation of the differentially expressed genes identified during the analysis. Since these extracellular vesicles likely contain different proteins, performing a proteomic analysis of the isolated exosomes would provide valuable insights.

- The authors claim that treatment with exosomes of both endothelial cells and cardiomyocytes protects against ischemia and reperfusion injury in vitro. However, their analysis primarily involves descriptive assessments of cell morphology to distinguish healthy from damaged cells, without quantification, and an LDH assay to assess cytotoxicity. It is strongly recommended to use a more specific method to analyze cell death, such as TdT staining, and quantify the extent of damage. Furthermore, evaluation of caspase pathway activation could provide additional insights.

- The manuscript would benefit significantly from including in vivo analyzes of the protective effect of these exosomes in a mouse model of cardiac ischemia and reperfusion injury.

- The discussion section needs extensive revision. Currently, it resembles a summary of the literature and the results of the study. It lacks a detailed commentary on the observed results or suggestions on the molecular mechanisms underlying these effects, which are missing throughout the manuscript.

For these reasons, I recommend rejecting the article in its current form.

Comments on the Quality of English Language

English could be improved. 

Author Response

Reviewer 1:

Comments and Suggestions for Authors

The current study investigates the potential cardioprotective role of exosomes from mesenchymal stromal cells (MSCs) cultured under different conditions on both endothelial cells and iPSC-derived cardiomyocytes against ischemia and reoxygenation injury. Exploring extracellular vesicles with specific recognition of cardiac cells for therapeutic purposes is of great interest and may represent an innovative and more effective approach to alleviate cardiovascular diseases. While the manuscript is intriguing and generally well written, there are some important comments and issues that require further attention.

- The RNAseq experiment primarily characterizes the content of exosomes obtained from MSCs cultured with different coatings, but lacks validation of the differentially expressed genes identified during the analysis. Since these extracellular vesicles likely contain different proteins, performing a proteomic analysis of the isolated exosomes would provide valuable insights.

Response: In this study, we used MSCs derived from 3 patients with heart failure who were enrolled in the FOCUS-CCTRN trial and who presented with improved clinical function (Morrissey et al. Cells 2022;11(13):2092 and Taylor et al. Cell Transplant 2016;25(9):1675-1687). The samples obtained for the ancillary study were used to generate the LAMA2-EXOs. The clinical trial expired, and the samples were discarded, eliminating the possibility of using additional samples for validation of our findings.

We agree that the analysis of exosomes' proteomics would provide valuable insights and would highlight the potential of microvesicle therapies. In this manuscript, our aim was to assess whether the specific ECM protein LAMA2 could alter the content of exosomes. We opted for mRNA sequencing since this technique is less variable than proteomics due to the massive protein translational regulation (Lu et al. Sci Data 2023;10:455; Schwanhäusser et al. Nature 2011;473: 337–342). In addition, most of the protein and RNA exosome content studies have been conducted on cells with a high proliferative capacity and a different phenotype from the MSCs used in our study to generate the EXOs.  

- The authors claim that treatment with exosomes of both endothelial cells and cardiomyocytes protects against ischemia and reperfusion injury in vitro. However, their analysis primarily involves descriptive assessments of cell morphology to distinguish healthy from damaged cells, without quantification, and an LDH assay to assess cytotoxicity. It is strongly recommended to use a more specific method to analyze cell death, such as TdT staining, and quantify the extent of damage. Furthermore, evaluation of caspase pathway activation could provide additional insights.

Response: In this study, we used LDH to indicate cell membrane integrity. As per the reviewer’s comment, to provide additional insight into cell damage in our experiments, we have now included new data from studies on caspase 3/7 activity. Our results showed injury patterns similar to those we found in our LDH studies. These data are now included in the Results section (pages 9 and 10, lines 232-237, and Fig 7C and D). In addition, we have renamed that section in the Methods as “Cell Damage” to better indicate the focus of those experiments.

- The manuscript would benefit significantly from including in vivo analyzes of the protective effect of these exosomes in a mouse model of cardiac ischemia and reperfusion injury.

Response: We agree with the reviewer that our study would benefit significantly from in vivo examination of the protective effects of exosomes and will be the next step in advancing this work. However, ours is a preliminary study, and in vivo analyses are beyond the scope of this current work. Nevertheless, we believe our early findings here could benefit the field of cardiovascular therapy with exosomes.

- The discussion section needs extensive revision. Currently, it resembles a summary of the literature and the results of the study. It lacks a detailed commentary on the observed results or suggestions on the molecular mechanisms underlying these effects, which are missing throughout the manuscript.

Response: We appreciate the reviewer’s insightful comment. We have now substantially revised the Discussion to include more commentary on the results and suggestions of underlying mechanisms. We believe this revision has improved the quality of the text.

For these reasons, I recommend rejecting the article in its current form.

Response: We respect the reviewer’s opinion; however, we believe that the additional experiments, analyses, and text revisions included in our new version have improved the quality and importance of our manuscript. We hope you now find our manuscript more suitable for publication.

Comments on the Quality of English Language

English could be improved.

Response: Our manuscript has been formally edited by a professional English-language editor.

Reviewer 2 Report

Comments and Suggestions for Authors

In this study, the Authors investigated the potential of mesenchymal stromal cell (MSC)-derived exosomes, particularly those cultured under laminin alpha 2 (LAMA2) coating, to enhance cardiomyocyte protection during ischemia-reoxygenation injury. The study mapped LAMA2-EXO and gelatin-EXO mRNA abundance, identifying differentially expressed genes associated with cardiac muscle development and extracellular matrix organization. The results indicate a significant protective effect of LAMA2-EXO on cardiomyocytes, leading to faster recovery compared to controls after ischemia-reoxygenation injury.

This study introduces a novel approach by utilizing MSC-derived exosomes cultivated under LAMA2 coating, targeting specific extracellular matrix proteins to enhance cardiomyocyte protection during ischemic injury. This approach may offer new insights into therapeutic strategies for ischemic heart disease. Therefore this well written and thoroughly done study can be published in IJMS. There are only several small recommendations that the Authors can easily address.

Abstract: You wrote: LAMA2-EXO had a 2-fold protective effect compared to non-treatment; after 1.5 hr of recovery (20% O2, 5% CO2, cardiomyocyte-enriched media), we found that cardiomyocytes treated with LAMA2-EXO recovered faster than the control group.   Please, specify how did you define a 2-fold protective effect?  In what units did you measure it?

Results:

1. I did not see the Supplementary Tables and Figures. Please, provide.

2. Please, provide figure 2 of better quality. The current version does not show gene symbols and therefor is not informative.

3. Please, provide references for gene functional descriptions and for the other descriptions where it is appropriate. For example, in the place where you write about the interaction between SH3BGRL3 and EGFR (lines 129-130)

4. Methods. Please provide more detailed description of differentially expressed gene analysis.

5. What threshold for FDR abd for gene expression difference did you use?

6. Please, describe software that you used for data visualization.

Author Response

Reviewer 2:

Comments and Suggestions for Authors

In this study, the Authors investigated the potential of mesenchymal stromal cell (MSC)-derived exosomes, particularly those cultured under laminin alpha 2 (LAMA2) coating, to enhance cardiomyocyte protection during ischemia-reoxygenation injury. The study mapped LAMA2-EXO and gelatin-EXO mRNA abundance, identifying differentially expressed genes associated with cardiac muscle development and extracellular matrix organization. The results indicate a significant protective effect of LAMA2-EXO on cardiomyocytes, leading to faster recovery compared to controls after ischemia-reoxygenation injury.

This study introduces a novel approach by utilizing MSC-derived exosomes cultivated under LAMA2 coating, targeting specific extracellular matrix proteins to enhance cardiomyocyte protection during ischemic injury. This approach may offer new insights into therapeutic strategies for ischemic heart disease. Therefore this well written and thoroughly done study can be published in IJMS. There are only several small recommendations that the Authors can easily address.

Response: We thank the reviewer for the positive feedback and kind words.

Abstract: You wrote: LAMA2-EXO had a 2-fold protective effect compared to non-treatment; after 1.5 hr of recovery (20% O2, 5% CO2, cardiomyocyte-enriched media), we found that cardiomyocytes treated with LAMA2-EXO recovered faster than the control group.   Please, specify how did you define a 2-fold protective effect?  In what units did you measure it?

Response: We appreciate the reviewer’s comment and apologize for any confusion. We have now updated the abstract to indicate that the 2-fold protective effect is related to plasma membrane integrity and the apoptosis activation pathway. The units used were fold change of arbitrary unit versus baseline and fold change of relative luminescence unit versus baseline. This information is also included in Figure 7.

Results:

  1. I did not see the Supplementary Tables and Figures. Please, provide.

Response: We apologize for any missing files. We were unable to verify on submission if the supplemental files were properly uploaded. We will ensure they are available on resubmission.

  1. Please, provide figure 2 of better quality. The current version does not show gene symbols and therefor is not informative.

Response: We apologize for the poor quality of Figure 2. To address this issue, we have now separated the original Figure 2 into two separate figures and provided higher quality images. Please see the new Figures 2 and 3.

  1. Please, provide references for gene functional descriptions and for the other descriptions where it is appropriate. For example, in the place where you write about the interaction between SH3BGRL3 and EGFR (lines 129-130)

Response: We thank the reviewer for highlighting the missing references. We have now included the pertinent references (new references 18-23) for each gene function description in the text.

  1. Methods. Please provide a more detailed description of differentially expressed gene analysis.

Response: As per the comment, we have now included additional details regarding the differential gene expression analysis in the Methods section. Please see Section 4.4 on page 13.

  1. What threshold for FDR abd for gene expression difference did you use?

Response: We apologize for the missing information. The FDR-adjusted p-value cut-off was set at 0.05, and the gene expression differences were calculated based on log-fold changes of 0.5. We have now included this information in the Methods section of the manuscript. Please see the bottom of Section 4.4.

  1. Please, describe software that you used for data visualization.

Response: The information regarding the camera and software used to capture brightfield images has now been included in the Methods section of the manuscript. Please see the end of Section 4.1. In addition, we now include information regarding the plot visualization for gene expression data in the Methods at the end of Section 4.4.

Round 2

Reviewer 1 Report

Comments and Suggestions for Authors

I appreciate the authors' efforts to improve the quality of the manuscript, but they only added one experiment and basically did not address any of my other concerns.

I still feel that there is insufficient evidence to support the manuscript's conclusion.

Author Response

We thank the editor for the feedback. As per the comment, we have now included a limitations paragraph (page 12, paragraph 3) and reframed the final section of the Discussion (page 12, paragraph 2). We hope these revisions address the editor’s concerns.

Reviewer 2 Report

Comments and Suggestions for Authors

The Authors pefectly addressed all poits. 

Tha Article can be published. 

Author Response

(The authors gave the same response as above.)
